# “Unraveling EMILIN-1: A Multifunctional ECM Protein with Tumor-Suppressive Roles” Mechanistic Insights into Cancer Protection Through Signaling Modulation and Lymphangiogenesis Control

**DOI:** 10.3390/cells14130946

**Published:** 2025-06-20

**Authors:** Samanta Muzzin, Enrica Timis, Roberto Doliana, Maurizio Mongiat, Paola Spessotto

**Affiliations:** Molecular Oncology Unit, Centro di Riferimento Oncologico Aviano (CRO), Istituto di Ricovero e Cura a Carattere Scientifico (IRCCS), 33081 Aviano, Italy; samanta.muzzin@cro.it (S.M.); enrica.timis@cro.it (E.T.); rdoliana@cro.it (R.D.)

**Keywords:** extracellular matrix, tumor microenvironment, lymphatic vessels, inflammation, proteolytic remodeling, neutrophil elastase

## Abstract

EMILIN-1 (Elastin Microfibril Interface Located Protein 1) is an extracellular matrix homotrimeric glycoprotein belonging to the EMILIN/Multimerin family, with both structural and regulatory roles, increasingly recognized for its tumor-suppressive functions. Initially identified for its involvement in elastogenesis and vascular homeostasis, EMILIN-1 has gradually emerged as a key player in cancer biology. It exerts its anti-tumor activity through both direct and indirect mechanisms: by regulating tumor cell proliferation and survival and by modulating lymphangiogenesis and the associated inflammatory microenvironment. At the molecular level, EMILIN-1 inhibits pro-oncogenic signaling pathways, such as ERK/AKT and TGF-β, via its selective interaction with α4/α9 integrins. In the tumor microenvironment, it contributes to tissue homeostasis by restraining aberrant lymphatic vessel formation, a process closely linked to tumor dissemination and immune modulation. Notably, EMILIN-1 expression is frequently reduced or its structure altered by proteolytic degradation in advanced cancers, correlating with disease progression and poor prognosis. This review summarizes the current knowledge on EMILIN-1 in cancer, focusing on its dual function as an active extracellular matrix regulator of intercellular signaling. Particular attention is given to its mechanistic role in the control of cell proliferation, underscoring its potential as a novel biomarker and therapeutic target in oncology.

## 1. Introduction

Multiple lines of evidence indicate that the extracellular matrix (ECM) plays a pivotal role in promoting tumor progression, with many of its components and cell receptors contributing to cancer cell development, migration, and dissemination. It is well established, for example, that integrin engagement positively influences cell proliferation [1]. In contrast, only a limited number of ECM proteins are known to exhibit intrinsic tumor-suppressive properties [2,3]. Among them, the glycoprotein EMILIN-1 stands out for its unique anti-proliferative function. Originally characterized for its involvement in elastogenesis and vascular homeostasis and the proper functioning of lymphatic vessels (LVs), EMILIN-1 was later described to modulate cell growth through specific interactions with α4β1 and α9β1 integrins.

Extensive research has demonstrated that this regulatory activity defines EMILIN-1 as a non-conventional ECM component with tumor-inhibitory capacity across various cancer types. EMILIN-1 actively participates in signaling pathways that restrain malignant behavior and suppress aberrant lymphangiogenesis—a key path for metastatic dissemination. In fact, its mRNA downregulation or proteolytic inactivation creates a microenvironment conducive to tumor expansion and immune escape.

This review consolidates current knowledge on EMILIN-1′s protective functions and highlights its potential as both a biomarker and a therapeutic target in oncology.

## 2. EMILIN-1: Structure, Expression and Physiological Functions

EMILIN-1 (Elastin Microfibril Interface Located Protein 1) is a homotrimeric glycoprotein [4] composed of an N-terminal EMI domain, a central coiled-coil region, and a C-terminal globular C1q-like (gC1q) domain. These domains mediate interactions with ECM components [5,6] and cell surface receptors (notably α4β1 and α9β1) [7,8], contributing to tissue organization and signaling pathways. EMILIN-1 is primarily localized at the interface between elastin and microfibrils in elastic tissues [4,9,10,11]. It is abundantly expressed in the walls of large blood vessels, skin, lungs, and other connective tissues where elastic fibers are essential for tissue resilience. The presence in the lymphatic system is particularly important for its biological function. It is specifically localized in the anchoring filaments of initial lymphatic capillaries and in the valves of lymphatic collectors [12,13] (Figure 1). Through its structural and signaling roles, EMILIN-1 is essential for maintaining LV integrity and for the proper function of the lymphatic vasculature.

EMILIN-1 displays strong adhesive and migratory properties for different cell types (fibroblasts, keratinocytes, trophoblast, hematopoietic, sarcoma and endothelial cells) [7,8,14]. The interaction between α4β1 and the gC1q domain of EMILIN-1 is particularly efficient and peculiar, implying the simultaneous involvement of all the three acidic residues of the trimeric ligand in the formation of a dimeric complex with the integrin βI domain [7,15]. Integrin α9β1 is expressed on keratinocytes, epithelial cell lines and also fibroblasts and is highly homologous to α4 [7,16]. The lack of stress fibers and focal adhesions in cells attached to EMILIN-1 indicates that cells are preferentially stimulated to migrate rather than to adhere firmly. The promigratory properties of EMILIN-1 have been demonstrated for several cell types [14,17], especially in non-pathological conditions: the finding that trophoblast cells attach and very efficiently migrate and haptotactically move on EMILIN-1 is particularly important in the first phases of uterine wall invasion process by trophoblasts [14]. Moreover, a cooperation of MMPs with α4β1 integrin has been suggested in this process: MMP14 and MMP2 are upregulated in co-cultures of trophoblast cells and stromal cells expressing EMILIN-1 and enhance their haptotaxis process towards EMILIN-1 [14].

## 3. Mechanisms of Tumor Suppression by EMILIN-1

### 3.1. Inhibition of Proliferation and Survival

As a secreted ECM glycoprotein primarily produced by stromal and mesenchymal cells, including fibroblasts, smooth muscle cells, and lymphatic endothelial cells, EMILIN-1 exerts its biological functions in the extracellular space, highlighting the need to interpret its role in cancer within the context of its stromal origin and ECM localization.

Besides the functional significance of adhesion and migration as the consequence of the interaction between EMILIN-1 and α4/α9, the striking aspect of this ligand/receptor engagement is related to proliferation. Targeted inactivation of the *Emilin1* gene in the mouse induces three phenotypes characterized by (1) systemic hypertension [18], (2) lymphatic alterations resulting in mild lymphedema [12], and (3) increased thickness of epidermis and dermis [8]. The lack of integrin occupancy by EMILIN-1, as occurs in *Emilin1*^−/−^ mice, leads to an increased number of Ki67-positive cells in the epidermis and dermis [8] and in the lymphatic valves [13]. The molecular mechanism underlying the regulatory role of EMILIN-1 in the skin has been well defined, providing evidence that PTEN plays a central role in the crosstalk between α4/α9β1 integrin and TGF-β signal pathways. EMILIN-1 binding to α4β1/α9β1 integrins empowers the downregulation of proliferative cues induced by TGF-β through the upregulation of PTEN and the consequent inhibition of ERK [8]. Moreover, the α4β1 integrin engagement triggers a series of events leading to the ubiquitination and degradation of HRasGTP, as demonstrated in a colon cancer model, that is responsible for pERK1/2 downregulation and low proliferation [19]. This evidence links the anti-proliferative capability of α4β1 integrin to a proto-oncogene like HRas through the switching off of its activated form (Figure 2).

To date, only Tissino et al. have demonstrated that EMILIN-1 triggers pro-survival signaling in chronic lymphocytic leukemia (CLL) cells through its specific interaction with the α4β1 integrin, as evidenced by enhanced ERK and AKT phosphorylation and reduced apoptosis in-vitro [20]. No other studies have corroborated this mechanism, suggesting that the effect may be context-dependent and influenced by the tumor microenvironment (TME).

Notably, EMILIN-1 exhibits unique functional properties: unlike the binding of other ECM ligands to α4 or α9 [21,22,23,24], the signal transmitted by EMILIN-1 has an anti-proliferative effect. Recent studies have shown that EMILIN-1 exerts its tumor suppressor function in head and neck squamous cell carcinomas by downregulating the cell cycle and Aurora kinase signaling pathways. However, this study does not provide supporting evidence for the involvement of integrins in this mechanism [25]. Qi et al. demonstrated that EMILIN-1 exerts anti-tumor activity in gastric cancer cells by upregulating TSPAN9 expression [26]. Mechanistically, TSPAN9 suppresses the secretion of metastasis-associated proteins, such as MMP9, through inhibition of the ERK1/2 pathway, thereby impairing gastric cancer cell proliferation, migration and invasion [27]. While the precise mechanism underlying EMILIN-1-mediated TSPAN9 regulation remains unclear, integrins likely facilitate cells-EMILIN-1 interactions that subsequently enhance TSPAN9 expression. Supporting this notion, Sun et al. demonstrated that ARID1 upregulates EMILIN-1 expression through direct promoter binding and that EMILIN-1 suppresses liver cancer cell migration and invasion in both in vitro and in vivo models [28]. Furthermore, activation of the transcription factor RORB upregulates EMILIN-1 expression, resulting in inhibition of metastasis, a finding that further confirms the protective role of this ECM molecule also in colorectal cancer [29].

### 3.2. Lessons from In Vivo Models

Collectively, these studies demonstrate that EMILIN-1 functions beyond its structural role, actively regulating both cell proliferation and metastatic dissemination. This tumor-suppressive capacity suggests that EMILIN-1-deficient microenvironments may foster tumor development. Supporting this hypothesis, Emilin1^−/−^ mice developed larger lymphangiomas compared to wild-type controls [12] and showed accelerated tumor growth, along with increased burden and size in a skin carcinogenesis model [30]. In addition, the increased LV density observed in both tumors and draining lymph nodes (LNs) of Emilin1^−/−^ mice was attributed to the absence of EMILIN-1′s anti-proliferative signaling mediated by the interaction with integrins α4β1/α9β1 [30]. This pro-tumorigenic microenvironment was further evidenced by the enhanced growth of α4β1 integrin-expressing B16F10 melanoma cells under EMILIN-1-deficient conditions [30]. The protective role of EMILIN-1 was further validated in a HER2-driven breast cancer model by crossing Emilin1^−/−^ mice with MMTV-Δ16HER2 transgenic mice that spontaneously develop multifocal mammary adenocarcinomas. Δ16HER2/Emilin1^−/−^ female mice showed accelerated mammary gland development, significantly earlier tumor onset, and increased tumor multiplicity compared to wild-type controls [31].

In the colon, the EMILIN-1 tumor-suppressive mechanism during chemically-induced carcinogenesis involves the suppression of AKT and ERK activation. EMILIN-1 deficiency results in increased tumor burden, particularly high-grade adenomas, and enhanced tumor growth. The critical role of α4β1/α9β1 integrin interaction was conclusively demonstrated using the E1-E955A knock-in mouse model (previously referred to as E933A, which did not include the protein’s signal peptide in the amino acid numbering), where the mutation of the gC1q domain specifically abrogates EMILIN-1′s binding to these integrins while preserving TGF-β regulation through the EMI domain [18,32]. This model further revealed the domain-specific functions of EMILIN-1: the gC1q-mediated integrin interaction exerted direct anti-proliferative effects, as evidenced by enhanced growth of YTN16 gastric cells in E1-E955A mice [33]. These findings establish that EMILIN-1′s tumor suppressor activity in gastrointestinal cancers operates through two distinct mechanisms: (1) integrin-dependent growth regulation via the gC1q domain and (2) TGF-β modulation through the EMI domain. Notably, wild-type mice developed only limited lesions following N-Methyl-N-nitrosourea (MNU) exposure, whereas E1-E955A mutants exhibited significantly increased lesion burden, particularly gastric intraepithelial neoplasia (GIN), a more aggressive lesion type characterized by elevated DNA damage and Ki-67 proliferation marker levels compared to conventional adenomas [33]. These findings illuminate EMILIN-1′s critical role in early gastric cancer development. The E1-E955A model represents a valuable preclinical tool, particularly given the growing recognition of ECM components as potential prognostic biomarkers in gastric cancer and the current paucity of animal models specifically designed to study microenvironmental and ECM alterations in carcinogenesis.

### 3.3. Inactivation of EMILIN-1 by Proteolytic Activity

The functional characterization of EMILIN-1 in proliferation and tumor suppression has been largely achieved through the Emilin1^−/−^ and E1-E955A mouse models. This raises an important translational question: do human pathophysiological conditions exist that mimic EMILIN-1 deficiency or functional impairment? While few pathogenic EMILIN-1 mutations have been identified in humans, those reported occur in regions outside N- and C-terminus domains and, therefore, do not affect its tumor regulatory properties [34,35].

However, emerging evidence suggests that proteolytic degradation may represent a clinically relevant mechanism of EMILIN-1 inactivation. In vitro studies demonstrate that EMILIN-1 is susceptible to cleavage by neutrophil elastase [36,37]. This finding is of particular significance given that neutrophil elastase actively influences cancer growth and development [38,39], neutrophil infiltration is a hallmark of many tumor microenvironments (TMEs) [40,41], and elevated neutrophil presence correlates with disease progression in gastric, renal and lung carcinomas [42,43,44,45]. Pivetta et al. provided direct evidence for this mechanism in human cancers, showing that neutrophil elastase-mediated cleavage impairs EMILIN-1′s tumor suppressor function and that EMILIN-1 degradation occurs in sarcomas and ovarian cancer specimens [37]. These findings position proteolytic degradation as a potentially critical pathway for EMILIN-1 inactivation during inflammatory processes and tumor progression. Moreover, researchers have documented the presence of neutrophil extracellular traps (NETs), structures decorated with granular elastase and myeloperoxidase that form during neutrophil activation [46]. This discovery provided direct evidence of neutrophil activation in the TME and localized elastase release. Notably, EMILIN-1 co-localized with these NET structures, demonstrating their spacial association in tumors [37]. Among other proteolytic enzymes, neutrophil elastase stands unique in its ability to specifically cleave the EMILIN-1′s regulatory highly compact gC1q domain. While matrix metalloproteinases like MMP3, MMP9, and MMP14 can partially degrade EMILIN-1, they fail to process the recombinant gC1q domain, which also shows resistance to collagenase and proteinase-3 (PR3) [36].

This proteolytic regulation extends to melanoma progression. Amor-Lòpez et al. identified a novel metastasis-promoting mechanism whereby EMILIN-1 gets proteolyzed and packaged into small extracellular vesicles (sEVs) in LN metastatic cells [47]. This process reduces intracellular EMILIN-1 levels, may represent an early metastatic adaptation, and parrales known tumor-suppressor elimination pathways [48]. Intriguingly, EMILIN-1′s vesicular trafficking may have context-dependent roles. Proteomic analyses reveal elevated EMILIN-1 levels in serum-derived EVs from mucinous colon adenocarcinomas (a clinically aggressive subtype) [49], suggesting either compensatory tumor-suppressor mobilization or pathologic co-option of vesicular transport. These findings position proteolytic degradation and vesicular secretion as complementary mechanisms of EMILIN-1 regulation across cancer types, with potential diagnostic and therapeutic implications.

LC–ESI–MS/MS analysis, combining high-performance liquid chromatography resolution with mass spectrometer accuracy, detected elevated levels of EMILIN-1 peptides among other plasma proteins in breast cancer patients [50]. This enrichment likely reflects tumor-associated proteolytic processing of EMILIN-1, consistent with the inactivation mechanisms observed in solid tumors, as described above.

Overall, these studies on EMILIN-1′s tumor-suppressive mechanisms highlight its role not only as a non–cell-autonomous regulator within the TME but also as an active modulator of cellular behavior. As an ECM protein, EMILIN-1 shapes the microenvironment through its deposition in the ECM and, crucially, exerts direct effects on cells via integrin-mediated signaling. This dual function is supported by our in vivo evidence showing that EMILIN-1 loss or degradation promotes tumor progression primarily through disruption of the surrounding microenvironment while also interfering with integrin-dependent cellular signaling pathways—underscoring its integrated role in maintaining tissue homeostasis and restraining tumor growth.

## 4. EMILIN-1 Expression in Tumors

EMILIN-1 has emerged as a promising biomarker in oncology, supported by its frequent identification in cancer-related gene expression signatures. However, its expression patterns exhibit tissue-specific variability that does not always correlate with its established tumor suppressor functions. This contextual duality is exemplified by several key findings. For instance, in non-small lung cancer stroma, EMILIN-1 upregulation correlated with low proliferation, contrasting with other stromal markers such as PDLIM5, SPARC and TAGLN that are associated with high proliferation rates [51]. Moreover, breast tumors responsive to doxorubicin showed elevated EMILIN-1 expression, suggesting a potential protective role in the TME [52], although this pattern did not reach significance in validation cohorts. Furthermore, a stemness-based radiosensitivity signature identified high EMILIN-1 expression as a marker of radiation resistance [53]. The potential role of EMILIN-1 in breast cancer radioresistance may stem from its complex interactions with ECM components and immune-modulatory functions within the TME, highlighting the complexity of its roles. These apparently contradictory roles may reflect tissue-specific regulations of EMILIN-1 function and/or differential post-translational modification patterns. Recent integrated single-cell RNA sequencing and spatial transcriptomic analyses of breast tumors revealed EMILIN-1 as a top differentially expressed gene in TGF-β-rich zones associated with CD8^+^ T-cell infiltration. Specifically, elevated EMILIN-1 expression at the tumor margins correlated with enhanced cytotoxic T-cell recruitment, attenuation of TGF-β-mediated immunosuppression and improved patient prognosis [54]. These findings highlight EMILIN-1′s immunomodulatory capacity in shaping an anti-tumor immune response.

The relationship between EMILIN-1 expression and clinical outcomes in mesenchymal tumors remains context-dependent. A tumor-suppressive role has been observed upon epigenetic silencing via promoter hypermethylation, which reduces EMILIN-1 expression in fusion-positive rhabdomyosarcoma [55] and uterine carcinosarcoma, particularly in carcinomatous components [56]. On the other hand, a potential tumor-promoting role has been proposed in osteosarcoma, where more elevated EMILIN-1 levels were detected compared to the less aggressive desmoids tumors [57], and in Ewing’s sarcoma [58,59], which also display elevated EMILIN-1 expression. In Ewing’s sarcoma, concomitant MMP9 upregulation suggests that proteolytic regulation may determine net EMILIN-1 bioavailability [58].

As mentioned, the functional role of EMILIN-1 in cancer appears to be highly context-dependent, with both tumor-suppressive and tumor-promoting activities observed across different malignancies. Proteomic analyses have identified EMILIN-1 upregulation in ovarian carcinomas [60], yet our findings demonstrate that the protein is frequently fragmented in ovarian cancer, leiomyosarcoma, and undifferentiated sarcoma specimens [37]. This suggests that while EMILIN-1 gene expression may be elevated, proteolytic processing by tumor or microenvironment-derived enzymes could render the protein non-functional. Such post-translational modifications may ultimately be more biologically significant than transcriptional regulation in determining EMILIN-1′s functional status within the TME.

In gastric cancer, conflicting reports exist regarding the EMILIN-1 role. While some bioinformatics analyses identify EMILIN1 as a part of a poor-prognostic cancer-associated fibroblast signature [61] and show a positive correlation with tumor stage and grade [62], other studies using the TNMplot tool revealed significantly decreased EMILIN-1 expression in gastric tumors compared to normal tissue [33]. This discrepancy may be explained by concomitant upregulation of MMP9 and MMP14 in gastric cancer [33], where increased MMP14 expression correlates with metastasis and poor prognosis [63]. Similar patterns of EMILIN-1 downregulation are observed in bladder, lung, ovarian, rectal, and uterine carcinomas [33], supporting its potential tumor-suppressive function across multiple cancer types.

The association between EMILIN-1 expression and brain tumor progression remains particularly controversial. In vitro models suggest low EMILIN-1 levels correlate with aggressive phenotypes [64], while bioinformatics analyses paradoxically link high EMILIN-1 expression to poor prognosis in low-grade gliomas [65]. A preliminary machine learning study identified EMILIN-1 as a potential therapeutic target in neuroblastoma [66], though these findings require validation in larger cohorts.

More consistent evidence supports EMILIN-1’s protective role in head and neck squamous cell carcinoma (HNSCC), where reduced expression correlates with second primary malignancies [67] and poor prognosis in HPV-associated cases [68].

These findings further underscore that EMILIN-1 plays a highly context-dependent role in cancer, with marked variability across different tumor types. Although EMILIN-1 frequently appears in cancer-related gene expression signatures and is often characterized as a tumor suppressor, its expression does not consistently correlate with malignant progression. This is especially apparent in mesenchymal tumors, where EMILIN-1 levels are heterogeneous and show limited predictive value for clinical outcomes. EMILIN-1 expression may appear upregulated in certain tumors; however, its tumor-suppressive activity depends on its functional integrity and interaction with specific integrins. Proteolytic degradation within the TME can impair this activity, suggesting that apparent expression does not necessarily reflect functional tumor suppression. Such discrepancies suggest that EMILIN-1’s functional relevance is influenced not only by the tumor cell type but also by the composition of the TME and stromal interactions. As an ECM component whose function is strongly modulated by post-translational modifications, the biological activity of EMILIN-1 cannot be inferred solely from transcriptomic data. Comprehensive assessments should account for the expression and activity of regulatory elements such as proteases and miRNAs, which critically shape its net functional output. Therefore, EMILIN-1 should not be regarded as a universal hallmark of tumor biology but rather as a context-specific regulator whose significance varies with tissue type, cellular milieu, and potential tumor stage. Acknowledging this variability is crucial for a proper understanding of its biomarker potential and therapeutic value.

## 5. EMILIN-1 and Lymphangiogenesis: A Central Role in Malignancy Control

Lymphangiogenesis, i.e., the formation of new LVs, establishes critical conduits for metastatic spread to LNs and distant organs [69,70]. This process involves dynamic remodeling of the lymphatic network, where both lymphangiogenesis and lymphatic hyperplasia enhance tumor cell intravasation into the lymphatic vasculature [71,72]. The metastatic cascade is further facilitated by structural adaptations in lymphatic collectors near sentinel LNs, which may involve the proliferation of lymphatic endothelial cells, non-proliferative lymphatic expansion mechanisms, and functional alterations in lymph-associated vascular smooth muscle cells [71,73]. Notably, lymphangiogenesis within tumor-draining LNs creates a permissive microenvironment that accelerates systemic dissemination, representing a pivotal step in metastatic dissemination.

### 5.1. Regulation of Lymphangiogenesis

EMILIN-1 occupies a unique position within the EMILIN/Multimerin family as a sole member demonstrating essential functions in lymphangiogenesis [12,13,32,74]. Initial characterization of EMILIN-1 knockout mice revealed its critical role in blood pressure homeostasis through EMI domain-mediated sequestration of proTGF-β. In *Emilin1*^−/−^ mice, uncontrolled TGF-β activation leads to three principal cardiovascular phenotypes: reduced vascular cell proliferation, decreased arterial lumen diameter, and elevated peripheral vascular resistance [18]. Beyond this canonical mechanism, EMILIN-1’s gC1q domain contributes to vascular homeostasis by modulating smooth muscle cell proliferation and arterial wall architecture through integrin-mediated signaling [75].

The protein’s vascular functions extend significantly to lymphatic biology, where it maintains structural integrity through several mechanisms. EMILIN-1 deficiency causes characteristic lymphatic abnormalities, including loss of anchoring filaments, vessel hyperplasia, and irregular patterning [12]. These structural defects correlate with functional impairments such as mild lymphedema, reduced lymph transport capacity, and increased vascular leakage [12]. The molecular basis involves specific α9β1 integrin interactions that are essential for proper lymphatic valve formation and maintenance [13], representing the first documented case of lymphatic dysfunction caused by an ECM protein deficiency. The E1-E955A mouse model, which carries a point mutation disrupting gC1q-integrin binding, has provided crucial mechanistic insights. These mutants exhibit abnormal lymphatic networks characterized by increased vessel density, tortuosity, and reduced anchoring filaments, along with malformed collector valves [74]. In vitro and ex vivo functional assays demonstrate that lymphatic endothelial cells from these mutants have severely impaired sprouting capacity and cannot form proper capillary structures. At the molecular level, the gC1q domain activates lymphangiogenic pathways through AKT and ERK phosphorylation in lymphatic endothelial cells, mirroring the effects of VEGF-C stimulation [32]. Notably, while both EMILIN-1 and VEGF-C promote lymphangiogenesis, they operate through distinct mechanisms. Unlike the established crosstalk between α5β1 integrin and VEGF receptors [76,77], gC1q-mediated α9β1 integrin activation does not induce VEGFR3 tyrosine phosphorylation. However, combined stimulation with gC1q and VEGF-C produces synergistic pathway activation, suggesting that optimal lymphatic development requires coordinated input from both integrin and growth factor receptor signaling systems [32] (Figure 3).

These collective findings have significantly advanced our understanding of EMILIN-1’s multifaceted role in lymphangiogenesis regulation, offering promising translational potential for treating lymphangiogenesis-related pathologies. A seminal study by Pivetta et al. using a post-surgical tail lymphedema mouse model revealed that the acute phase of acquired lymphedema coincides with neutrophil elastase-mediated EMILIN-1 proteolysis [78]. This enzymatic degradation leads to compromised lymphatic endothelial cell junctions and severely impaired drainage to regional LNs. Therapeutic intervention with sivelestat, a selective neutrophil elastase inhibitor [79], effectively preserved EMILIN-1 integrity and restored lymphatic function, demonstrating a significant reduction in lymphedema. The clinical relevance of these findings was further strengthened by the detection of characteristic elastase-cleaved EMILIN-1 fragments in human secondary lymphedema specimens [78], providing compelling evidence for neutrophil elastase inhibition as a viable treatment strategy.

### 5.2. Impact on Tumor Metastasis and Inflammatory Landscape

EMILIN-1 has been consistently demonstrated to function as a metastasis suppressor in multiple experimental models. Comparative studies in both chemically-induced skin carcinogenesis and syngeneic tumor transplantation models (using luciferase-expressing B16F10 and LLC cell lines) revealed significantly increased LN metastasis in *Emilin1*^−/−^ mice compared to wild-type controls [30]. This prometastatic phenotype likely stems from two complementary mechanisms. First, the structural defects in lymphatic endothelial cells characteristic of EMILIN-1 deficiency, including reduced anchoring filaments and impaired intercellular junctions [12], create physical gaps that facilitate tumor cell intravasation. Second, the enhanced lymphangiogenesis observed in both primary tumors and sentinel LNs of EMILIN-1 mutant mice [30,33,74] establishes an expanded network of conduits for tumor cell dissemination. Together, these findings position EMILIN-1 as a critical regulator of the metastatic cascade through its dual roles in maintaining lymphatic vascular integrity and suppressing pathological lymphangiogenesis.

EMILIN-1 serves as a critical regulator in the TME, maintaining LV integrity while preventing pathological lymphangiogenesis. Its functional importance is particularly evident in inflammatory contexts, where impaired lymphangiogenesis—normally essential for inflammation resolution and immune cell clearance—can perpetuate chronic inflammation that promotes malignant progression [80,81]. This relationship is exemplified in the colon, where EMILIN-1 deficiency exacerbates tumor development through combined effects of lymphatic dysfunction and chronic inflammation [32]. The gastric microenvironment presents a unique scenario where inflammation, lymphangiogenesis, and ECM remodeling collectively drive pathogenesis [82,83,84]. Here, EMILIN-1 appears to function as a multifunctional sentinel by preserving LV integrity, exerting direct anti-proliferative effects, and suppressing oncogenic progression. The use of animal models revealed a striking correlation between aberrant LV morphology and increased gastric malignancy [33]. Transgenic studies demonstrate that EMILIN-1 deficiency leads to excessive tumor-associated lymphangiogenesis, accelerated gastric cancer cell dissemination via LVs, and more aggressive lesion development. These findings suggest that EMILIN-1 normally constrains gastric cancer progression by modulating the lymphatic network’s structure and function. The observed reduction of EMILIN-1 in human samples correlates with both LV abnormalities and increased inflammatory infiltrates, likely due to impaired immune cell clearance through dysfunctional LVs [33].

Through its dual structural and signaling roles in the ECM and lymphatic vasculature, EMILIN-1 serves as a crucial regulator of inflammatory processes. The protein maintains proper LV function, enabling effective interstitial fluid drainage and immune cell trafficking. When EMILIN-1 is deficient, impaired lymphatic function leads to pathological fluid retention and establishes a chronic pro-inflammatory microenvironment [32,33,85].

The consequences of EMILIN-1 deficiency extend to myeloid cell accumulation, with notable increases in tissue-resident monocytes and macrophages. This phenomenon arises through multiple mechanisms: (1) compromised lymphatic clearance of immune cells, (2) altered secretion of inflammatory chemokines and cytokines, and (3) dysregulated TGF-β signaling—a key pathway controlling macrophage polarization and immune suppression [85]. Together, these effects disrupt normal immune homeostasis, promoting a persistent inflammatory state that can fuel disease progression.

## 6. Therapeutic and Diagnostic Implications

Although EMILIN-1 does not fit the classical definition of a tumor suppressor based on genetic inactivation, its tumor-suppressive properties arise from non–cell-autonomous mechanisms, primarily through modulation of the tumor microenvironment. Its frequent downregulation or proteolytic inactivation in various cancers—often correlating with advanced disease and poor outcomes—highlights its potential as a context-dependent biomarker. Quantitative evaluation of EMILIN-1 levels or its cleavage products in tissues and biofluids may offer valuable tools for diagnosis, prognosis, and disease monitoring.

From a therapeutic perspective, several strategies targeting EMILIN-1 pathways show considerable promise. Functional restoration approaches may include gene therapy to reintroduce EMILIN-1 expression or administration of recombinant protein. Alternative interventions could focus on preventing its degradation through protease inhibitors or developing peptide mimetics that retain its integrin-binding and signaling functions. These approaches may prove particularly effective against tumors with prominent lymphatic dissemination when combined with conventional therapies.

The role of EMILIN-1 in modulating the TME suggests additional opportunities to enhance immunotherapy efficacy. By restoring stromal integrity and immune homeostasis, EMILIN-1-based interventions could potentially overcome the current limitations of immunotherapeutic approaches. Future research efforts should prioritize clinical validation of EMILIN-1 as a theranostic target, coupled with translational studies to evaluate its therapeutic potential in combination with existing treatment modalities, as summarized in Table 1.

## 7. Conclusions and Future Directions

EMILIN-1 has emerged as a pleiotropic ECM glycoprotein whose tumor-suppressive functions substantially transcend its canonical structural role. Operating at the critical interface between cellular signaling and tissue organization, EMILIN-1 orchestrates a dual anticancer mechanism: directly inhibiting neoplastic cell proliferation and survival while simultaneously remodeling the TME through precise regulation of lymphangiogenesis and inflammatory responses.

The molecular basis of EMILIN-1’s protective effects involves sophisticated modulation of oncogenic signaling cascades, particularly through ERK/AKT pathway inhibition and TGF-β homeostasis, mediated by its specific interaction with α4β1/α9β1 integrin receptors. Its capacity to constrain pathological LV formation establishes a formidable barrier against metastatic spread, particularly through lymphatic routes. The clinical relevance of these mechanisms is underscored by the consistent finding of EMILIN-1 downregulation or proteolytic inactivation in advanced malignancies, correlating with aggressive disease phenotypes and poor prognostic outcomes.

While these discoveries have significantly advanced our understanding, critical knowledge gaps persist. The molecular regulators governing EMILIN-1 expression dynamics and degradation patterns in malignant transformation remain incompletely characterized. Moreover, the translational potential of EMILIN-1 as either a diagnostic biomarker or therapeutic target demands systematic investigation through rigorous preclinical validation and clinical studies.

As an underappreciated modulator of tumor biology, EMILIN-1 presents exciting opportunities for innovative cancer interventions. Elucidating its multifaceted roles may unlock novel therapeutic paradigms focused on ECM normalization and metastasis prevention, potentially offering new avenues for combination therapies in advanced malignancies.

## Figures and Tables

**Figure 1 cells-14-00946-f001:**
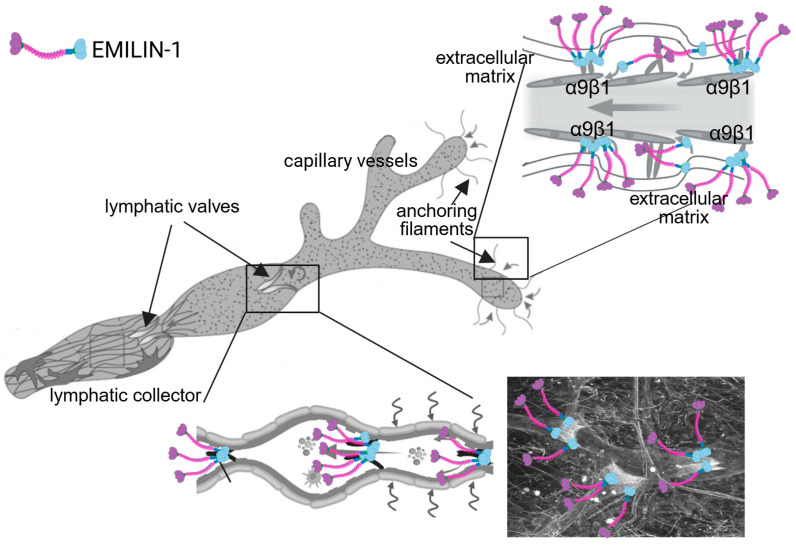
Localization of EMILIN-1 in the lymphatic vasculature. Schematic representations and whole-mount immunofluorescence staining illustrate the distribution of EMILIN-1 (colored icons) in the lymphatic vasculature. In capillaries, EMILIN-1 decorates fibrillar structures (anchoring filaments) connecting endothelial cells to the surrounding ECM, contributing to vessel stability and function. In collecting vessels, EMILIN-1 is enriched at valve leaflets, where it may play a role in maintaining valve integrity and unidirectional lymph flow (grey arrows). The immunofluorescence image was obtained using whole-mount staining of mature lymphatic valves of adult mouse ear skin [13]. The entire valve matrix core is positive for EMILIN1, with the staining (grey signal) concentrated on the free edge of the leaflet fibers.

**Figure 2 cells-14-00946-f002:**
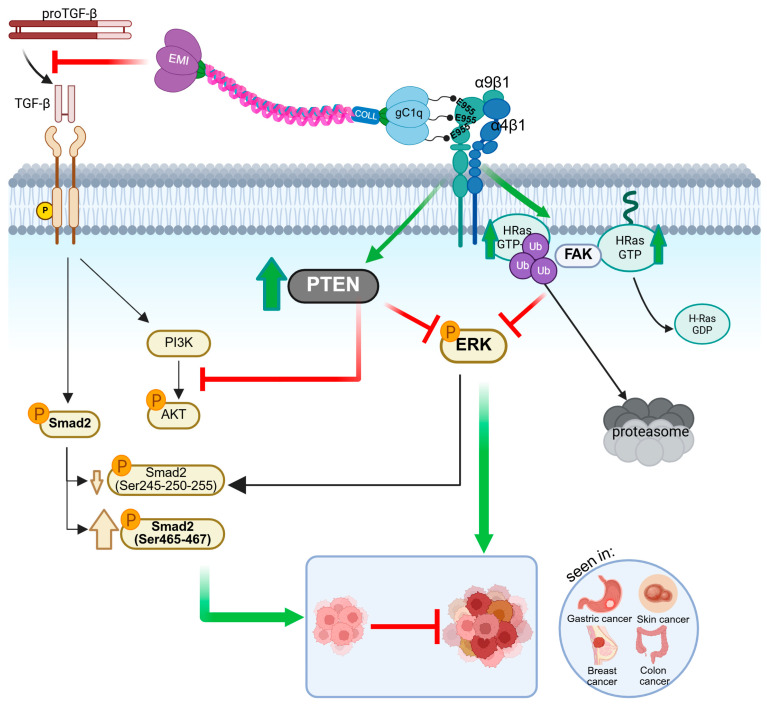
EMILIN-1 and proliferation. The illustration depicts the molecular mechanisms through which EMILIN-1 regulates cell proliferation. Under physiological conditions, TGF-β exerts a cytostatic effect primarily by inducing Smad2 phosphorylation at Ser465/467 and modulating the PI3K/Akt pathway by regulating PTEN expression. EMILIN-1 disrupts this signaling by binding specifically to the pro-TGF-β precursor, thereby inhibiting its extracellular maturation. In addition, EMILIN-1 interacts with α4β1 and α9β1 integrins via the E955 (E954 in humans) residue of each of its monomers, promoting PTEN activation and suppressing the pro-proliferative ERK1/2 pathway. Through this dual mechanism, EMILIN-1 counteracts TGF-β–driven proliferative signaling. In the absence of EMILIN-1, the unregulated maturation of TGF-β leads to increased levels of its active form, while α4/α9β1 integrins are no longer engaged, resulting in PTEN downregulation. This contributes to enhanced activation of proliferative signaling pathways, including pAkt and pERK1/2. Additionally, activation of α4β1 integrin by the gC1q domain stimulates FAK, leading to an increase in HRasGTP levels and its subsequent ubiquitination (HRasGTP-Ub). HRasGTP inactivation, together with low basal intracellular Ca^2+^ levels, contributes to decreased ERK1/2 phosphorylation and reduced cell proliferation. Created with BioRender.com.

**Figure 3 cells-14-00946-f003:**
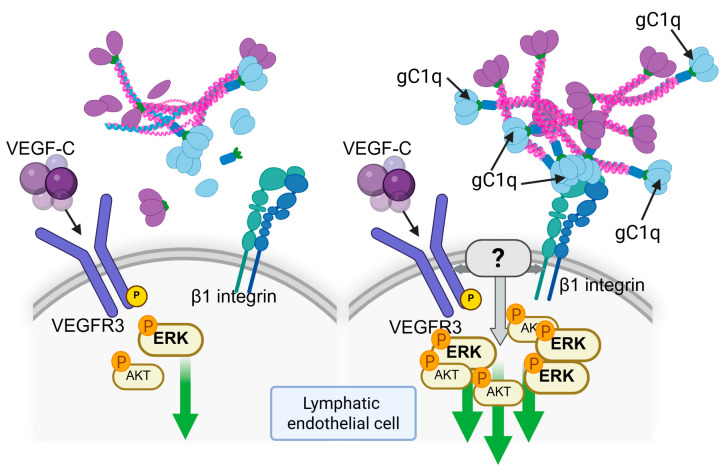
Schematic representation of the cooperative signaling between gC1q and VEGF-C in lymphatic endothelial cells. The gC1q domain binds to integrin α9β1, while VEGF-C activates VEGFR3. In the absence of gC1q–integrin engagement (e.g., when EMILIN-1 is degraded by proteases), VEGF-C/VEGFR3 signaling alone supports lymphatic endothelial cell survival (left panel). Although gC1q by itself does not induce VEGFR3 phosphorylation, its binding to integrins enhances and prolongs downstream signaling cascades—such as AKT and ERK—when VEGF-C is present. This indicates a yet incompletely understood mechanism (grey arrows) in which integrin and VEGFR3 signaling pathways synergize to promote effective lymphangiogenic responses. Receptor interactions and signaling events are based on previously published data [32,74]. Green arrows indicate the activaction of downstream signaling. Created with BioRender.com.

**Table 1 cells-14-00946-t001:** Therapeutic and diagnostic implications.

Application Area	Implication	Notes/Examples
Diagnostic Biomarker	Reduced EMILIN-1 expression or proteolysis correlates with tumor aggressiveness	Potential use in tumor grading, prognosis, and monitoring
Prognostic Marker	Loss of EMILIN-1 associated with poor clinical outcomes	Especially relevant in cancers with lymphatic dissemination
Therapeutic Target	Restoration of expression/function	Gene therapy, recombinant protein delivery, or ECM-targeted therapies
Protease Inhibition	Preventing EMILIN-1 degradation	Use of MMP or neutrophil elastase inhibitors to preserve EMILIN-1 structure
Peptide Mimetics	Design of EMILIN-1-derived peptides	Mimicking integrin-binding domains to block pro-oncogenic signaling
Pathway Modulation	Targeting EMILIN-1-regulated pathways (e.g., TGF-β, VEGF-C, ERK/AKT)	Possible synergy with targeted therapies or immunotherapies
TME Modulation	Rebalancing TME via lymphangiogenesis control	May reduce metastasis and improve immune cell infiltration

## Data Availability

Not applicable.

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
