# Peer review of "“Unraveling EMILIN-1: A Multifunctional ECM Protein with Tumor-Suppressive Roles” Mechanistic Insights into Cancer Protection Through Signaling Modulation and Lymphangiogenesis Control"

_cells, 2025, doi:10.3390/cells14130946_

Round 1
Reviewer 1 Report
Comments and Suggestions for Authors
The manuscript is clear, and the information is complete and well-explained/organized.
There are very few suggestions I can make to the authors; for example, check the abbreviations throughout the text to be sure they are explained (e.g., LV on page 2, TME and LN page 6).
In paragraph 2 the description of EMILIN-1 is summarized, but only on page 12 is reported that it's a glucoprotein, and nothing is written about the glucidic moiety.
Description of figure 1 should be more specific, in particular, regarding what it concerns the immunofluorescence image
The last paragraph of chapter 2 requires some additional information about the types of cells described.
Page 4, line 134: the sentence starting with "furthermore---" should be rewritten
Figure 3 description in the text is extensively analyzed, but the image itself is scanty: should the authors add an arrow to indicate gC1q domain? Since the figure reports 2 situations, the caption should be more descriptive with particular regard to the left or the right part of the illustration.
Author Response
The manuscript is clear, and the information is complete and well-explained/organized.
We greatly appreciate the Reviewer’s constructive feedback, which provided us with the opportunity to improve and clarify several aspects of the manuscript.
There are very few suggestions I can make to the authors; for example, check the abbreviations throughout the text to be sure they are explained (e.g., LV on page 2, TME and LN page 6).
- We have reviewed all abbreviations in the manuscript and made the necessary corrections accordingly.
In paragraph 2 the description of EMILIN-1 is summarized, but only on page 12 is reported that it's a glucoprotein, and nothing is written about the glucidic moiety.
- We have added to paragraph 2 that EMILIN-1 is a homotrimeric glycoprotein, and we included the reference to in vitro studies showing that the protein is highly glycosylated. However, the exact composition of its sugar moiety has not been extensively characterized. We believe that this level of detail would not significantly contribute to the central aims of the manuscript.
Description of figure 1 should be more specific, in particular, regarding what it concerns the immunofluorescence image
- We have revised the figure legend to include the following sentence:
“The immunofluorescence image was obtained using whole-mount staining of mature lymphatic valves from adult mouse ear skin [11]. The entire valve matrix core is positive for EMILIN-1, with the staining (grey signal) concentrated on the free edge of the leaflet fibers.”
Additionally, we have specified the colored icon used in the figure to indicate the localization of EMILIN-1 within lymphatic structures, including anchoring filaments and valves.
The last paragraph of chapter 2 requires some additional information about the types of cells described.
- We have added the following information at the beginning of the paragraph: (fibroblasts, keratinocytes, trophoblasts, hematopoietic, sarcoma, and endothelial cells).
Page 4, line 134: the sentence starting with "furthermore---" should be rewritten
- The sentence in question has been rewritten
Figure 3 description in the text is extensively analyzed, but the image itself is scanty: should the authors add an arrow to indicate gC1q domain? Since the figure reports 2 situations, the caption should be more descriptive with particular regard to the left or the right part of the illustration.
- We thank the Reviewer for the helpful suggestion. We have slightly modified the figure by adding arrows to clearly illustrate the potential synergistic effect of gC1q engagement with its integrin. The figure legend has been updated accordingly, and the description of the illustration has been improved to reflect these changes.

Reviewer 2 Report
Comments and Suggestions for Authors
The authors primarily describe findings from several in vitro experiments, essentially based on cell lines, including 18 publications of their research group from the past 20 years (= relatively high self-citation level). And as expected, there is a lot of focus on the functions of EMILIN1 in endothelial cells and blood or lymph vessel formation, which is understandable and also justified.
A brief examination of EMILIN1 expression across human normal tissues and tumors reveals that it is highly tissue-specific. It is very highly and consistently expressed in lymph endothelial cells across many tissues, and this is a major topic in the review. But it is even more consistently expressed in a few subtypes of mesenchymal cells, including fibroblasts (stromal cells in tumors), and smooth muscle cells - these levels appear higher in general than in the endothelial cells. Epithelial cells, however, rarely express any of it. And yet, there is not much talk about the role/function of EMILIN1 in these stromal/mesenchymal cell types, especially in cancer initiation and progression. There is consistent, but low-level, expression in macrophages and a few other immune cells. (Data from single-cell sequencing as publicly available in Proteinatlas). Similarly, immunohistochemistry IHC staining of EMILIN1 across a few epithelial tumor types, like had & neck squamous cell carcinomas (HNSCC) or other squamous and adenocarcinoma cancer types confirm that EMILIN seems indeed mainly expressed in the stromal compartment (CAFs and smooth muscle cells), and lymph endothelial cells/lymph vessels, but doesnt show up in epithelial cancer cells in squamous carcinoma nor adenocarcinomas. There is very little or no expression in the actual epithelial tumor cells. (again, evaluating stainings and scRNAseq data from proteinatlas and other sources).
With this surprisingly specific expression pattern in mind, I wonder how many of the functions assigned to EMILIN1 in the review and in the papers cited in this manuscript could be explained? How does a gene that's not expressed in the tumor cells downregulate the cell cycle and Aurora kinase signaling pathways? (Ref. 23, line 123). Or act on tumor cells by upregulating TSPAN9 expression in gastric cancer cells (Ref. 24, line 126). How can "EMILIN1 suppress liver cancer cell migration and invasion in both in vitro and in vivo models ?" (line 133). It just eludes me, and should be clarified.
In general, I am not questioning these and many other findings referred to within the manuscript, but the authors could put a bit more emphasis on the putative mechanisms HOW EMILIN1 might be involved in any of these processes, especially when its not just cell line-based experiments, but one of the many animal models that are also referred to here. Is it cell-cell-interaction that is critical (via the integrin connection?) and not so much the EMILIN1 in the tumor cells themselves, rather the EMILIN expression on cells OTHER than the tumor cells?
I know the authors are aware of this issue, in principle, but its not coming across in the manuscript as a major mechanism; although I believe this is a very hot topic these days: what is the role of the full cellular complexity & communication in the tumor microenvironment (TME), how do tumor/CAF and tumor/endothelial cell interactions drive progression? Maybe EMILIN1 is a major communicator protein.
For example, a typical finding (probably correctly) referred to in the manuscript reads like the following: "EMILIN-1 deficiency results in increased tumor burden, particularly high-grade adenomas, and enhanced tumor growth. The critical role of α4β1/α9β1 integrin interaction was conclusively demonstrated using the E1-E955A knock-in mouse model E933A" (line 173 - 176). How could all of this be explained if it's true that the protein is NOT expressed much (or not at all) in the tumor cells themselves, but mainly/mostly in the stroma? I'm not alluding that any of the observations referred to in the manuscript are wrong in any way - they just don't seem to align with the very high tissue- and cell-type specificity I observe across several databases.
If this question were addressed specifically in the review, like discussing how a gene that's not expressed in the tumors can still affect tumor progression, it would only make the manuscript MORE interesting. I consider this a very relevant question.
Furthermore, the authors cite several papers in which the EMILIN1 function is described as that of a tumor suppressor. That may be the case, I don't argue here (but can't believe it's true). But this statement may again be a bit difficult to explain or support, following the observation that expression of EMILIN1 is UP in several tumor types and not down. For example, this seems to be the case in head and neck cancers, for both mRNA and protein expression, and protein is confirmed to be exclusively expressed in the stromal compartment, not in the tumor.
This issue is critical, and it is briefly mentioned in lines 242 - 245: "EMILIN-1 has emerged as a promising biomarker in oncology, supported by its frequent identification in cancer-related gene expression signatures. However, its expression patterns exhibit tissue-specific variability that doesn't always correlate with its established tumor suppressor functions". The rest of the discussion in this chapter, however, often refers to tumors in a general way and as if EMILIN1 is a core feature of the tumor cells.
Which may be in mesenchymal tumors (lines 265 - 274). The authors mention that the "relationship between EMILIN-1 expression and clinical outcomes in mesenchymal tumors remains context-dependent" (line 265), and that is maybe one of the core features of this gene, across many or most tumor entities? In principle, I think the authors should just stress and elaborate on this issue much more, to do this interesting gene full justice.
Chapter 5 on lymphangiogenesis doesn't have any such bias issues and is a clear expertise of the authors, too.
However, chapter 6 seems to generalize again: "The tumor-suppressive properties of EMILIN-1 offer significant potential for both diagnostic and therapeutic applications in cancer management" (lines 426-427). From what I see across multiple open-access databases (cBioportal, GEPIA2, TISCH2, etc), I cannot simply agree with the statement above. In most cancer types, EMILIN1 appears not to be prognostic for patient survival (both overall and disease-free survival). There are typically no mutations observed for the gene in most tumor types, as it is often the case for genetic drivers of cancer initiation or progression (for example, check NOTCH1 in HNSCC); and one would expect loss-of-function mutations at last in some tumors, but they aren't there. Or they are rare, like in HNSCC (only 8 missense mutations across 530 tumors). Doesn't look like a strong tumor suppressor to me.
Again, this is not meant as a criticism to discredit the manuscript's validity, on the contrary, it should be seen as "constructive criticism" and as a suggestion to discuss these points and improve the relevance of the paper.
On the very positive side, I like the Figures 1-3, they are very informative, clear, and well done as illustrations for the review's main topics.
Another issue that could be briefly mentioned: I noticed there are also EIMILIN2 and EMILIN3 in the human genome. Are they in any way functionally related to what EMILIN1 is doing in cancers, or not relevant here?
Author Response
We are grateful to the Reviewer for the careful reading of our manuscript and for the valuable suggestions and comments. These have been very helpful in refining the manuscript and strengthening our interpretations.
The authors primarily describe findings from several in vitro experiments, essentially based on cell lines, including 18 publications of their research group from the past 20 years (= relatively high self-citation level). And as expected, there is a lot of focus on the functions of EMILIN1 in endothelial cells and blood or lymph vessel formation, which is understandable and also justified.
- We thank the Reviewer for this observation. We would like to point out that EMILIN-1 is a niche ECM protein whose functional roles—particularly in the context of endothelial biology—have been largely uncovered by our research group over the past two decades. As such, the current literature on EMILIN-1 remains relatively limited, and many of the foundational and mechanistic insights referenced in our manuscript necessarily draw from our own previously published work. While we have made a concerted effort to include and acknowledge relevant contributions from other research groups where available, the specialized nature of this protein has inevitably required a degree of self-citation to ensure a thorough and accurate representation of its biological functions. Nonetheless, we have supplemented the manuscript with nine additional references pertaining to broader topics beyond EMILIN-1 to further balance the citation landscape.
A brief examination of EMILIN1 expression across human normal tissues and tumors reveals that it is highly tissue-specific. It is very highly and consistently expressed in lymph endothelial cells across many tissues, and this is a major topic in the review. But it is even more consistently expressed in a few subtypes of mesenchymal cells, including fibroblasts (stromal cells in tumors), and smooth muscle cells - these levels appear higher in general than in the endothelial cells. Epithelial cells, however, rarely express any of it. And yet, there is not much talk about the role/function of EMILIN1 in these stromal/mesenchymal cell types, especially in cancer initiation and progression. There is consistent, but low-level, expression in macrophages and a few other immune cells. (Data from single-cell sequencing as publicly available in Proteinatlas). Similarly, immunohistochemistry IHC staining of EMILIN1 across a few epithelial tumor types, like had & neck squamous cell carcinomas (HNSCC) or other squamous and adenocarcinoma cancer types confirm that EMILIN seems indeed mainly expressed in the stromal compartment (CAFs and smooth muscle cells), and lymph endothelial cells/lymph vessels, but doesnt show up in epithelial cancer cells in squamous carcinoma nor adenocarcinomas. There is very little or no expression in the actual epithelial tumor cells. (again, evaluating stainings and scRNAseq data from proteinatlas and other sources).
- We thank the Reviewer for this detailed and thoughtful comment. We fully agree that EMILIN-1 is primarily expressed by stromal and mesenchymal cell types—particularly fibroblasts, smooth muscle cells, and lymphatic endothelial cells—and is rarely detected in epithelial cells. This is entirely consistent with both published data and our own observations. We would like to emphasize that EMILIN-1 is a secreted ECM glycoprotein, and its biological activity is mainly exerted in the extracellular space, where it modulates cell behavior through interactions with the microenvironment. Therefore, evaluating EMILIN-1 function solely based on mRNA expression—particularly in single-cell RNA-seq datasets—may be misleading, as transcript levels do not necessarily reflect the presence, amount, or distribution of the secreted protein within the tissue. For this reason, functional roles of EMILIN-1 must be interpreted in the context of its deposition and activity in the ECM, which is largely shaped by its stromal/mesenchymal sources. We acknowledge that the role of EMILIN-1 in stromal compartments of tumors—such as cancer-associated fibroblasts (CAFs) and the tumor microenvironment—deserves further exploration, especially in the context of cancer initiation and progression. In this review, our focus was primarily on endothelial and lymphatic biology, where the functional mechanisms of EMILIN-1 have been most clearly defined to date. Nevertheless, we agree with the Reviewer that its expression and potential function in mesenchymal/stromal compartments represent an important and emerging area of research, and we have added a comment to this effect at the beginning of Chapter 3 in the revised version of the manuscript.
With this surprisingly specific expression pattern in mind, I wonder how many of the functions assigned to EMILIN1 in the review and in the papers cited in this manuscript could be explained? How does a gene that's not expressed in the tumor cells downregulate the cell cycle and Aurora kinase signaling pathways? (Ref. 23, line 123). Or act on tumor cells by upregulating TSPAN9 expression in gastric cancer cells (Ref. 24, line 126). How can "EMILIN1 suppress liver cancer cell migration and invasion in both in vitro and in vivo models ?" (line 133). It just eludes me, and should be clarified.
- Most of our studies on the protective role of EMILIN-1 in cancer have demonstrated that its biological effects depend on its interaction with specific cell surface receptors—namely, integrins α4β1 or α9β1. The intracellular signaling events we observe are the result of a classical ligand–receptor interaction, in which EMILIN-1 functions as the ligand and integrins serve as its receptors. There is substantial literature supporting the concept that ECM components within the tumor microenvironment regulate the behavior of various cell types through interactions with different receptors. This is a well-established mechanism of ECM-mediated signaling. With regard to the references cited by the Reviewer (from other research groups), we share similar concerns about the lack of mechanistic detail provided. In fact, in reference 23, we specifically noted the absence of mechanistic insight, and in reference 24, we explicitly stated that “the precise mechanism underlying EMILIN-1-mediated TSPAN9 regulation remains unclear.” As for the findings on liver cancer published in Cancer Cell (Sun et al.), the effects were observed following EMILIN-1 overexpression, representing a forced experimental system. In this context, the results may reflect an autocrine regulatory mechanism, whereby cancer cells are artificially induced to secrete EMILIN-1, which then interacts with their own integrins to exert inhibitory effects.
In general, I am not questioning these and many other findings referred to within the manuscript, but the authors could put a bit more emphasis on the putative mechanisms HOW EMILIN1 might be involved in any of these processes, especially when its not just cell line-based experiments, but one of the many animal models that are also referred to here. Is it cell-cell-interaction that is critical (via the integrin connection?) and not so much the EMILIN1 in the tumor cells themselves, rather the EMILIN expression on cells OTHER than the tumor cells?
- We thank the Reviewer for raising this important point. We fully agree that a deeper emphasis on the putative mechanisms of EMILIN-1 action—particularly in the context of in vivo models—adds value to the discussion. As correctly noted, EMILIN-1 does not exert its effects through direct expression in tumor cells, but rather through its deposition in the ECM and interaction with specific integrins (such as α4β1 and α9β1) expressed not only on tumor, but also on stromal and endothelial cells. This interaction is key to modulating various signaling pathways that influence cell adhesion, migration, invasion and lymphangiogenesis. Thus, EMILIN-1 functions primarily as a non-cell-autonomous regulator within the TME. In our previous in vivo studies, we observed that the loss or degradation of EMILIN-1 in the ECM resulted in increased tumor growth and metastasis—not because of changes in tumor cells per se, but due to disrupted signaling in the surrounding stroma and vasculature.
We have now added at the end of Chapter 3 clarifying statements to the revised manuscript to more explicitly describe this mechanism of action and to highlight the relevance of EMILIN-1 expression by non-tumor cells in shaping tumor behavior.
I know the authors are aware of this issue, in principle, but its not coming across in the manuscript as a major mechanism; although I believe this is a very hot topic these days: what is the role of the full cellular complexity & communication in the tumor microenvironment (TME), how do tumor/CAF and tumor/endothelial cell interactions drive progression? Maybe EMILIN1 is a major communicator protein.
For example, a typical finding (probably correctly) referred to in the manuscript reads like the following: "EMILIN-1 deficiency results in increased tumor burden, particularly high-grade adenomas, and enhanced tumor growth. The critical role of α4β1/α9β1 integrin interaction was conclusively demonstrated using the E1-E955A knock-in mouse model E933A" (line 173 - 176). How could all of this be explained if it's true that the protein is NOT expressed much (or not at all) in the tumor cells themselves, but mainly/mostly in the stroma? I'm not alluding that any of the observations referred to in the manuscript are wrong in any way - they just don't seem to align with the very high tissue- and cell-type specificity I observe across several databases.
If this question were addressed specifically in the review, like discussing how a gene that's not expressed in the tumors can still affect tumor progression, it would only make the manuscript MORE interesting. I consider this a very relevant question.
- We thank the Reviewer for this insightful comment, which touches on a crucial and timely aspect of tumor biology—the complex and dynamic communication between cancer cells and the surrounding stromal and vascular compartments within the TME. We fully agree that EMILIN-1 should be viewed within this broader context of intercellular crosstalk. In fact, the effects we observe in our in vivo models—such as increased tumor burden and progression in EMILIN-1-KO or E1-E955A KI mice—are not attributable to direct expression of EMILIN-1 in tumor cells, but rather to its functional role as a matrix-bound signaling modulator expressed by stromal and endothelial cells. This underscores the importance of non-cell-autonomous tumor suppressive mechanisms. EMILIN-1 acts as a structural and functional ECM component that shapes cell behavior by engaging integrin-mediated signaling on surrounding malignant and non-malignant cells, such as fibroblasts, lymphatic and blood endothelial cells, and potentially immune cells. Its degradation or absence disrupts these signaling circuits, creating a permissive environment for tumor progression. In the revised manuscript, we have addressed this point more explicitly at the end of Chapter 3 by clarifying that the biological activity of EMILIN-1 depends on its presence in the ECM and its interaction with cell surface integrins.
Furthermore, the authors cite several papers in which the EMILIN1 function is described as that of a tumor suppressor. That may be the case, I don't argue here (but can't believe it's true). But this statement may again be a bit difficult to explain or support, following the observation that expression of EMILIN1 is UP in several tumor types and not down. For example, this seems to be the case in head and neck cancers, for both mRNA and protein expression, and protein is confirmed to be exclusively expressed in the stromal compartment, not in the tumor.
-We thank the Reviewer for raising this important point and for the opportunity to clarify our interpretation of EMILIN-1’s role as a tumor suppressor. We fully agree that the expression profile of EMILIN-1 can appear paradoxical, particularly when its levels are maintained or even increased in some tumor types, such as HNSCC. However, we believe that this apparent contradiction can be reconciled by considering the spatial and cellular compartmentalization of EMILIN-1 expression, and the fact that its tumor-suppressive activity is also mediated through stromal mechanisms. As also highlighted by the Reviewer, EMILIN-1 is largely absent in epithelial tumor cells and predominantly localized to the stromal compartment. In this context, its upregulation may represent a reactive stromal response rather than an oncogenic feature. However, despite its presence in the tumor stroma, EMILIN-1 can be functionally inactivated through proteolytic degradation by tumor- or stroma-derived proteases, as we and others have shown. This degradation impairs its ability to engage integrins and exert regulatory effects on cell adhesion, migration, and lymphangiogenesis—ultimately facilitating tumor progression. Therefore, the tumor-suppressive function of EMILIN-1 does not necessarily correlate with its bulk mRNA or protein levels in tumor tissue, but rather with its structural integrity and availability as a functional ECM component. We have now clarified this point at the end of Chapter 4 in the revised manuscript to better reflect the complex relationship between EMILIN-1 expression, localization, and functional activity.
This issue is critical, and it is briefly mentioned in lines 242 - 245: "EMILIN-1 has emerged as a promising biomarker in oncology, supported by its frequent identification in cancer-related gene expression signatures. However, its expression patterns exhibit tissue-specific variability that doesn't always correlate with its established tumor suppressor functions". The rest of the discussion in this chapter, however, often refers to tumors in a general way and as if EMILIN1 is a core feature of the tumor cells.
Which may be in mesenchymal tumors (lines 265 - 274). The authors mention that the "relationship between EMILIN-1 expression and clinical outcomes in mesenchymal tumors remains context-dependent" (line 265), and that is maybe one of the core features of this gene, across many or most tumor entities? In principle, I think the authors should just stress and elaborate on this issue much more, to do this interesting gene full justice.
- We fully agree that the context-dependent nature of EMILIN-1 expression and function—particularly the distinction between epithelial and mesenchymal tumors—requires further elaboration to avoid overgeneralization. In the revised version of the manuscript, we have added a dedicated paragraph to clarify that EMILIN-1 is not a universal hallmark of tumor cells, but rather a gene whose role in oncogenesis varies depending on tissue type, tumor histotype, and the microenvironment. We now emphasize at the end of Chapter 4 that while EMILIN-1 frequently appears in oncogenic gene signatures and exhibits tumor-suppressive activity in several models, its clinical significance and biological function remain heterogeneous. We believe this addition and changes in the previous manuscript improve the accuracy and depth of the discussion, and more accurately reflects the complex biology of EMILIN-1 across tumor entities.
Chapter 5 on lymphangiogenesis doesn't have any such bias issues and is a clear expertise of the authors, too.
However, chapter 6 seems to generalize again: "The tumor-suppressive properties of EMILIN-1 offer significant potential for both diagnostic and therapeutic applications in cancer management" (lines 426-427). From what I see across multiple open-access databases (cBioportal, GEPIA2, TISCH2, etc), I cannot simply agree with the statement above. In most cancer types, EMILIN1 appears not to be prognostic for patient survival (both overall and disease-free survival). There are typically no mutations observed for the gene in most tumor types, as it is often the case for genetic drivers of cancer initiation or progression (for example, check NOTCH1 in HNSCC); and one would expect loss-of-function mutations at last in some tumors, but they aren't there. Or they are rare, like in HNSCC (only 8 missense mutations across 530 tumors). Doesn't look like a strong tumor suppressor to me.
- We thank the Reviewer for the thoughtful and detailed comment. We agree that EMILIN-1 does not fulfill the classical definition of a tumor suppressor gene in terms of frequent inactivating mutations or consistent prognostic correlations across large cancer datasets. However, we would like to clarify that the tumor-suppressive role of EMILIN-1, as supported by multiple experimental models, arises from its extracellular functions—particularly its regulatory effects on the TME—rather than from intrinsic genetic alterations within cancer cells. Indeed, the concept of tumor suppression has evolved beyond cell-autonomous genetic events. A growing body of evidence supports the idea that certain proteins can act as non-cell-autonomous tumor suppressors by modulating the ECM, cell–ECM interactions, or the stromal–tumor interface. EMILIN-1 exerts its effects largely by interacting with specific integrins, thereby regulating processes such as lymphangiogenesis, cell proliferation, and tissue homeostasis—mechanisms that are crucial for tumor containment and immune surveillance. Although EMILIN-1 may not show frequent loss-of-function mutations or strong prognostic value across all cancer types in large datasets, its functional inactivation—for example, through proteolytic degradation in the TME—can still lead to tumor-promoting consequences. In this context, its tumor-suppressive properties are context-dependent and mediated by microenvironmental disruption, rather than gene mutations.
We have revised the statement at the beginning of Chapter 6 of the manuscript to better reflect this nuanced view and to avoid potential misinterpretation regarding the mechanism by which EMILIN-1 exerts its anti-tumor effects.
Again, this is not meant as a criticism to discredit the manuscript's validity, on the contrary, it should be seen as "constructive criticism" and as a suggestion to discuss these points and improve the relevance of the paper.
- We truly appreciate the intent to strengthen the manuscript by highlighting points that could improve its depth and relevance. We have carefully considered the suggestions and revised the text accordingly, where appropriate. Constructive input such as this is invaluable in refining the clarity and impact of our work, and we are grateful for the opportunity to address these aspects.
On the very positive side, I like the Figures 1-3, they are very informative, clear, and well done as illustrations for the review's main topics.
- We sincerely thank the Reviewer for the positive feedback regarding Figures 1–3. We are pleased to hear that the illustrations were found to be clear and informative, as they were specifically designed to visually support and clarify the main concepts discussed in the review. We have slightly modified Figure 3 as requested by Reviewer 1
Another issue that could be briefly mentioned: I noticed there are also EIMILIN2 and EMILIN3 in the human genome. Are they in any way functionally related to what EMILIN1 is doing in cancers, or not relevant here?
- Regarding the additional comment on EMILIN-2 and EMILIN-3, we appreciate the Reviewer’s suggestion. Indeed, EMILIN-2 and EMILIN-3 are part of the same protein family and share structural similarities with EMILIN-1. However, to date, their roles in cancer are much less well characterized. EMILIN-2 has been reported to be involved in apoptotic regulation and may exert pro-apoptotic functions in certain tumor contexts, while EMILIN-3 remains poorly studied and is mainly expressed during development. Given the limited data available and the lack of strong evidence linking these proteins to the mechanisms discussed in this review, we chose to focus on EMILIN-1.

Round 2
Reviewer 2 Report
Comments and Suggestions for Authors
Frankly, I am very pleased by the professional way the authors answer this reviewers slightly critical questions and how they have adressed the suggestions in the current 2nd version of the manuscript. I hope (and believe) it has increased in significance and is quite a thorough and informative recapitulation of what's known about EMILIN-1.
The authors have added several small additions to chapter 3 and a large one to chapter 4, and even added new references and changed one figure legend. Thats pretty much all we can ask for at this stage of the reviewing process and I wish the authors many citations from this pleasant to read manuscript/review article.